# Gene editing of *PKLR* gene in human hematopoietic progenitors through 5' and 3' UTR modified TALEN mRNA

Oscar Quintana-Bustamante[1,2‡]*, Sara Fañanas-Baquero[1,2‡], Israel Orman[1,2], Raul Torres[3,4], Philippe Duchateau[5], Laurent Poirot[5], Agnès Gouble[5], Juan A. Bueren[1,2], Jose C. Segovia[1,2]

1 Division of Hematopoietic Innovative Therapies, Centro de Investigaciones Energéticas Medioambientales y Tecnológicas/Centro de Investigación Biomédica en Red de Enfermedades Raras (CIEMAT/CIBERER), Madrid, Spain, 2 Instituto de Investigación Sanitaria Fundación Jiménez Díaz (IIS-FJD, UAM), Madrid, Spain, 3 Centro Nacional de Investigaciones Oncológicas (CNIO), Madrid, Spain, 4 Instituto Josep Carreras, Barcelona, Spain, 5 CELLECTIS, Paris, France

‡ These authors equally contributed to this work and should be considered as first authors on this work.
* oscar.quintana@ciemat.es

## Abstract

Pyruvate Kinase Deficiency (PKD) is a rare erythroid metabolic disease caused by mutations in the *PKLR* gene, which encodes the erythroid specific Pyruvate Kinase enzyme. Erythrocytes from PKD patients show an energetic imbalance and are susceptible to hemolysis. Gene editing of hematopoietic stem cells (HSCs) would provide a therapeutic benefit and improve safety of gene therapy approaches to treat PKD patients. In previous studies, we established a gene editing protocol that corrected the PKD phenotype of PKD-iPSC lines through a TALEN mediated homologous recombination strategy. With the goal of moving toward more clinically relevant stem cells, we aim at editing the *PKLR* gene in primary human hematopoietic progenitors and hematopoietic stem cells (HPSCs). After nucleofection of the gene editing tools and selection with puromycin, up to 96% colony forming units showed precise integration. However, a low yield of gene edited HPSCs was associated to the procedure. To reduce toxicity while increasing efficacy, we worked on i) optimizing gene editing tools and ii) defining optimal expansion and selection times. Different versions of specific nucleases (TALEN and CRISPR-Cas9) were compared. TALEN mRNAs with 5' and 3' added motifs to increase RNA stability were the most efficient nucleases to obtain high gene editing frequency and low toxicity. Shortening ex vivo manipulation did not reduce the efficiency of homologous recombination and preserved the hematopoietic progenitor potential of the nucleofected HPSCs. Lastly, a very low level of gene edited HPSCs were detected after engraftment in immunodeficient (NSG) mice. Overall, we showed that gene editing of the *PKLR* gene in HPSCs is feasible, although further improvements must to be done before the clinical use of the gene editing to correct PKD.

**Data Availability Statement:** All relevant data are within the paper and its Supporting Information files.

**Funding:** This work was supported by grants from "Ministerio de Economía, Comercio y Competitividad y Fondo Europeo de Desarrollo Regional (FEDER)" (SAF2017-84248-P), "Fondo de Investigaciones Sanitarias, Instituto de Salud Carlos III" (RD16/0011/0011) and "Instituto de Investigación Sanitaria de la Fundación Jiménez Díaz". The funders had no role in study design, data collection and analysis, decision to publish, or preparation of the manuscript.

**Competing interests:** PD, LP and AG are employee of Cellectis. JAB and JCS are consultants for Rocket Pharmaceuticals. This does not alter the authors' adherence to PLOS ONE policies on sharing data and materials.

# Introduction

Pyruvate kinase deficiency (PKD) is the most common erythroid inherited enzymatic defect causing chronic nonspherocytic hemolytic anemia. The prevalence of PKD is estimated at 51 cases per million among the white population[1]. PKD is an autosomal recessive disorder caused by mutations in the Pyruvate kinase L/R gene (*PKLR*), which lead to a total or partial reduction of the activity of the erythroid isoform of pyruvate kinase protein (R-type pyruvate kinase, RPK). To date, more than 200 different mutations in the *PKLR* gene have been linked to PKD[2]. This disease is associated with reticulocytosis, splenomegaly and iron overload, and may be life-threatening in severely affected patients. Therapy options for PKD are palliative and include regular red blood cell transfusion, splenectomy and iron chelation therapy. So far, allogeneic hematopoietic stem cell transplant (HSCT) represents the only curative treatment for severely affected patients[3]. However, allogeneic HSCT is not considered as routine in these patients and the overall survival rate after allogeneic HSCT in PKD is relatively low, mainly due to the toxicity of the procedure and graft versus host disease (GvHD)[3]. Moreover, the availability of compatible donors is very low. Autologous HSCT of genetically corrected cells would overcome these limitations[4].

Autologous HSCT of genetically corrected cells, also called hematopoietic stem cell (HSC) gene therapy is being used to treat many blood cell genetic diseases[4], including several red blood cell pathologies such as β-thalasemia[5, 6] or sickle cell disease[7]. We have recently developed a lentiviral vector to correct PKD[8], which has been granted orphan drug designation (EU/3/14/1330; FDA #DRU-2016-5168). This lentiviral-mediated gene therapy approach would offer a durable and curative clinical benefit with a single treatment.

Over the last few years, gene editing has emerged as a promising gene therapy approach for blood cell disorders, since genetic mutations can be accurately corrected. The swift spread of gene editing has been possible thanks to the development of engineered DNA nucleases such as zinc-fingers nucleases (ZFN), Transcription activator-like effector nucleases (TALEN) or cluster regularly interspaced palindromic repeats (CRISPR)-Cas9 system. Gene editing technology allows gene inactivation, integration at specific locus or correction of specific mutation. This promising technology has been adapted to correct hematopoietic inherited diseases as an ideal gene therapy. Different approaches have been reported as potential gene therapy for several blood cell diseases. Initially, these gene editing strategies were developed in induced Pluripotent Stem Cells (iPSCs) to correct different genetic pathologies such as X-linked chronic granulomatosis (X-CGD)[9], α-thalassemia[10, 11], β-thalassemia[10–12], sickle cell anemia [13, 14], Diamond Blackfan anemia[15] or Fanconi Anemia[16]. In this direction, we previously described the correction of PKD phenotype in patient-specific iPSC through a knock-in strategy mediated by a specific TALEN®, targeting the second intron of *PKLR* gene[17], establishing that gene editing may be a potential therapeutic option to correct PKD. Thanks to recent developments, gene editing is now considered as a real therapeutic option to be applied to hematopoietic stem and progenitor cells (HSPCs) directly. Different groups have reported efficient gene editing to correct genetic blood cell diseases such as X-linked severe combined immunodeficiency (SCID-X1)[18], JAK3 severe combined immunodeficiency[19], X-CGD [20], β-thalassemia[21–24], sickle cell anemia[21, 23–28] and Fanconi Anemia[29]. The high level of correction in HSPCs obtained without significant off-target effects suggests that the clinical use of gene editing therapy to correct genetic hematopoietic diseases is highly likely in the short term.

Here, we describe the rational path to adapt our previous gene editing approach to correct PKD in HSPCs, which is based on the optimization of the protocol conditions and the

inclusion of RNA stabilizer domains[30, 31], and show the advantages and limitations of current editing approaches for the treatment of PKD.

## Material and methods

### Cell lines and primary cells from healthy donors

HEK293T cells were obtained from ATCC and grown in DMEM (Thermo Fisher Scientific) supplemented with 10% Hyclone (GE Healthcare) and 1% penicillin/streptomycin solution (Gibco).

Umbilical cord blood samples (CB) from healthy donors were provided by *Centro de Transfusión de la Comunidad de Madrid*. All samples were collected under written consent and *Centro de Transfusión de la Comunidad de Madrid*'s institutional review board agreement (number PKDEFIN [SAF2017-84248-P]). Mononuclear cells were obtained by fractionation in Ficoll-hypaque according to manufacturer's recommendations (GE Healthcare). Purified CD34$^+$ cells were obtained using a MACS CD34 Micro-Bead kit (Miltenyi Biotec) and were kept frozen or used fresh in further experiments. Cells were viably frozen in 10% dimethyl sulfoxide solution and stored at–liquid nitrogen. Cells were grown in StemSpam (StemCell Technologies) supplemented with 0.5% penicillin/streptomycin solution (Gibco), SCF (100 ng/ml), TPO (100 ng/ml), Flt3 ligand (100 ng/ml) (all of them from EuroBiosciences). Cells were cultured at 37˚C, 5% $CO_2$ and 20% $O_2$.

### Gene editing tools

Donor matrix and plasmids expressing *PKLR* TALEN were described previously[32]. Briefly, donor matrix serves as template for homologous recombination in the intron 2 of the *PKLR* locus. The homology arms (HAs) were designed around a *PKLR* TALEN target sequence without containing it. Between the HAs the following elements were introduced: 1) Expression Cassette (EC), composed (going 5' to 3') of a splicing acceptor (SA), a codon optimized cDNA encoding RPK exons 3–11 with a FLAG tag and SV40 polyadenylation (polyA) signal. 2) Selection cassette (SC), composed of the mouse PGK promoter driving the expression of the Puromycin resistance and Tymidine Kinase fusion protein, and the bGH PolyA signal and flanked by LoxP sequences.

Each *PKLR* TALEN® plasmid carries one of the two subunits of specific TALENs targeting the second intron of *PKLR* gene (target sequence: `TGATCGAGCCACTGTACTCCAGCCTAG GTGACAGACGAGACCCTAGAGA`) under EF1α promoter. Additionally, these plasmids were modified to replace the constitutive expression promoter by T7 promoter for *in vitro* transcription of each *PKLR* TALEN® subunit, with or without either 5'UTR of the Venezuelan Equine Encephalitis Virus (VEEV)[30] or 3´UTR *β-Globin*. mRNA *PKLR* TALEN® subunits were *in vitro* transcribed and polyadenylated with mMESSAGE mMACHINE® T7 Ultra Kit (Thermofisher) according the manufacturer's instruction.

A single guide RNA (sgRNA) was identified with Zhang's software (http://crispr.mit.edu/) to target the same region targeted by *PKLR* TALEN®. *PKLR* sgRNA, targeting TAGGGTC TCGTCTGTCACCT sequence, was complementary to part of the spacer region and one of the *PKLR* TALEN® monomers. *PKLR* sgRNA was cloned in an all-in-one plasmid (kindly provided by Dr. Raul Torres, CNIO, Madrid, Spain), where *PKLR* sgRNA was expressed under the control of the U6 promoter and spCas9 under SFFV promoter. Additionally, RNP was made either by assembling tracrRNA, specific *PKLR* crRNA (6 or 12 μg) and spCas9 (3.3 or 6.6 μg) protein according IDT´s instruction (IDT or Supplier A), or by combination of *in vitro* synthesized guide RNA (6 or 12 μg) with a commercial Cas9 (4.5 or 9 μg) (PNA Bio or Supplier B) following incubation at room temperature for 10 minutes.

## Cell transfection and drug selection

DNA plasmid and/or *in vitro* synthesized mRNA were transfected into HEK293T cells with Lipofectamine 2000 (Thermofisher) according to manufacturer's instructions. After 7 days, HEK293T cells were cultured in presence of puromycin (1μg/ml, Sigma-Aldrich) for 14 days. For clonal analysis, 10 HEK293T cells were cultured in individual p96 wells for two weeks previous to their analysis.

Pre-stimulated CB-CD34[+] were electroporated with different amounts of nuclease DNA or mRNA and co-electroporated with the homologous recombination matrix by nucleofection according to the optimized protocol for AMAXA II Nucleofection system (Lonza, Basel, Switzerland). Between $8x10^5$-$1x10^6$ CB-CD34[+] cells were electroporated in cuvettes together with 5 μg of the recombination matrix and 2.5–5μg of nucleases using the U08 program. After nucleofection, cells were left in the cuvette for 20 minutes at 37˚C in order to recover from the electric pulse and then pipetted and transferred into the warmed culture medium. Then, electroporated CB-CD34[+] cells were cultured under the conditions previously described. After 4 or 6 days, [+] puromycin (1μg/ml) was added to the culture medium and cells were cultured for two or four additional days. Additionally, in some experiments 10μM dmPGE$_2$ (Cayman Chemical) was added in CB-CD34[+] culture after nucleofectionand maintained through the whole *in vitro* procedure in order to improve the survival and the engraftment capacity of edited CB-CD34[+] cells [33, 34].

## In vitro semisolid cultures

After puromycin selection, part of the culture was plated in methylcellulose (Myltenyi Biotech) and 14 days afterwards colonies were counted and scored based on their morphological appearance in a blinded fashion. Genomic DNA from puromycin resistant colonies was extracted according to the following protocol: single colonies were pelleted and resuspended in 10 μl of PBS. Genomic DNA was extracted using 20 μl of lysis buffer (0.3 mM Tris HCl pH 7.5, 0.6 mM CaCl$_2$, 1,5% Glycerol, 0.675% Tween-20 and 0.3 mg/ml Proteinase K) and incubated at 65˚C for 30 min, 90˚C for 10 min and 4˚C. After lysis, 30 μl of water was added, as previously described in Charrier et al[35].

## Nuclease activity assays

For Surveyor analysis, gDNA was extracted using NucleoSpin® Tissue kit (Macherey-Nagel). Then, a PCR was performed to amplify the homology region of the *PKLR-int2* locus in which the nucleases cut. The primers used in this PCR were PKLR-DS-3 Fw (`GGTAAATGGCAAA ACCCATC`) and PKLR-DS-3 Rv (`GGAAAGAAAGCAAGCAGGC`). The PCR was performed as follows: 95˚C 5min, 33 cycles of 1min at 95˚C, 45 s at 60˚C and 45 s at 72˚C, and one final step for 10 min at 72˚C. This PCR amplifies a 301 bp product that is then forced to form heteroduplex by the following cycle in the termocycler: 95˚C for 10 min, 95˚C to 85˚C (2˚C/ sec), 85˚C to 25˚C (0.1˚C/ sec), 4˚C forever. Then, 1 μl of Surveyor nuclease S and 1 μl of Surveyor enhancer S were added and were incubated for 1 hour at 42˚C. The digested products were evaluated by separation on a 10% Novex TBE gel with Novex TBE Running Buffer and Novex TBE Hi density Sample Buffer (Thermofisher). The samples were electrophoresed for 1.5 hours at 100 V and were stained with 1:10000 diluted SyberGold (Thermofisher). If an INDEL has occurred, the heteroduplex presents a DNA hairpin that can be recognized by the Surveyor® nuclease (Surveyor® mutation detection kit, IDT) and cut, generating a band pattern that could be visualized on 10% TBE Gels 1.0 mm (Thermofisher). Images from gels were analyzed in order to measure the cleavage fraction using Image J software (it measures the densitometry value of the different bands). The percentage of cleavage is determined by the

following equation:

$$\%NHEJ = \frac{Cleavaged\ bands - (2 \times Background)}{(Cleavaged\ bands + Parental\ band) - (3 \times Background)} \times 100$$

In order to determine nuclease activity in a more precise manner, INDEL frequencies were quantified using the TIDE software. PCR of genomic DNA extracted at 3 days after nucleofection according Macherey Nagel's instructions was performed and Sanger sequenced (Stabvida). Unedited cells were always used as a negative control for calculating INDEL frequencies with TIDE. The primers used to amplify the intron 2 region were the same used for Surveyor assay (PKLR-DS-3 Fw and PKLR-DS-3 Rv). Sanger sequencing was done with PKLR-DS-3 Fw primer.

## Gene targeting analysis by nested PCR

After puromycin selection, CB-CD34+ cells were plated in methylcellulose (Myltenyi), and 14 days afterwards colonies were counted and scored based on their morphological appearance in a blinded fashion to quantify hematopoietic progenitor potential. Genomic DNA from either single CFU or clonal HEK293T p96 well was extracted as previously described.

To detect the targeted integration of the Homology Directed Repair (HDR) cassette in the *PKLR* locus, we performed a Nested PCR, in which two different pairs of primers were used: *KI PKLR out 1F* (ACTGGGTGATTCTGGGTCTG) and *KI PKLR out 4R* (GGGGAACTTCCTGAC TAGGG) for the amplification of the LHA and the recombination cassette, obtaining a large amplicon of 3307bp. Then, 0.5μl of the previous PCR was used as template of a second PCR, where *KI PKLR in 3F* (GCTGCTGGGGACTAGACATC) and *KI PKLR in 1R* (CGCCAAATCTC AGGTCTCTC) primers were used to amplify a smaller region corresponding with the LHA (around 1982bp).

Both reactions were carried out by Herculase II Fusion High-Fidelity DNA Polymerase (Agilent Technologies) following a standard PCR protocol of 95°C 5min, 33 cycles of 1min at 95°C, 45 s at 58°C and 3min30s at 72°C, and one final step for 10 min at 72°C. Products were then visualized on a 1% agarose gel. For the assessment of the DNA fragments weight, two different DNA markers were used: DNA marker λ was made after digesting Phage Lambda (cI857 Sam 7) with BstE II restriction enzyme, and DNA marker IX was generated after digestion of ΦX174 plasmid DNA with Hae III.

## Transplantation of CB-CD34+ HSPCs into NSG mice

All animal experiments were performed in compliance with European and Spanish legislations and institutional guidelines. The protocol was approved by "Consejeria de Medio Ambiente y Ordenación del Territorio" (Protocol number PROEX 073/15). All surgery was performed under ketamine and Dex-medetomidina anesthesia, and all efforts were made to minimize suffering. Depending on the experiment, 1x10⁶ CB-CD34+ were electroporated and selected by puromycin four days later for other 48 hours. From 6x10⁴ to 1.45x10⁵ puromycin resistant hematopoietic cells were administered through tail vein to NSG (NOD.Cg-*Prkdc^scid Il2rg^tm1Wjl*/ SzJ) mice sublethally irradiated the day before transplant with 1.5 Gy. Either three or four NSG mice were transplanted with cells electroporated with donor matrix plus *PKLR* TALEN as DNA or *PKLR* as mRNA respectively in three independent experiments. One month after transplant, the animals were anesthetized and bone marrow (BM) aspiration was performed in order to analyze the human content by hCD45-FITC (BeckmanCoulter), hCD34-Pecy7 (eBioscience) and mCD45.1-PE (BD) staining by flow cytometry (LSR Fortessa; Becton Dickinson Pharmingen). Fluorochrome-matched isotypes were used as controls. 4',6-Diamidino-2-phenylindole (DAPI; Roche)-positive cells were excluded from the analysis. Analysis was performed using

FlowJo software. Four months after transplantation, mice were sacrificed by cervical dislocation and then BM of these animals was collected and stained to analyze the percentage of gene-targeted cells. Additionally, hCD45⁺/hCD34⁺ population was sorted in an Influx Cell Sorter (BD), cultured in presence of puromycin for two additional days and seeded in methylcellulose. After 14 days, CFUs were counted and picked individually to analyze for gene editing in the NSG engrafted hematopoietic progenitors.

## Statistical analysis

Statistical significance was determined using Kruskal-Wallis test with GraphPad Prism 7. The mean±SD and the individual values are represented in each graph. Additionally, P values are indicated in the graphs.

## Results

### Gene editing of the *PKLR* locus in human hematopoietic progenitor cells

In order to study the feasibility of inserting a therapeutic RPK cDNA into the homologous gene (knock-in gene editing approach) of human HSPCs, we adapted the strategy and tools previously described for the gene editing of PKD-hiPSC[17] to human cord blood hematopoietic progenitors (CB-CD34⁺). This gene editing strategy was based on the insertion of a codon-optimized version of the partial RPK cDNA (coRPK) in the second intron of the *PKLR* gene. The therapeutic cDNA construct covered exons 3 to 11, and contained a splicing adaptor at the 3′- end and an in-frame flag tag at the 5′-end, together with a constitutively expressed puromycin selection cassette (Fig 1A).

CB-CD34⁺ cells were stimulated for 24 hours before nucleofection with three different plasmids containing the therapeutic donor and the two subunits of the *PKLR* TALEN separately (Fig 1B). Electroporated cells were expanded for 6 days and selected with puromycin (1μg/ml of culture medium) for another 4 days. Semisolid cultures were performed at the end of the selection period for the identification of hematopoietic progenitors (colony forming unit [CFU] assay). A fraction of the electroporated cells was maintained without puromycin to estimate nucleofection and selection efficiencies.

Firstly, we observed a reduction in the total number of CFUs when the hematopoietic progenitors were electroporated with both *PKLR* TALEN® and the donor matrix (Fig 1C), showing the toxicity associated to the nucleofection of high amounts of DNA. This decrease in the number of CFUs was not detected when CB-CD34⁺ cells were sham electroporated (CTL) or electroporated with the matrix plasmid (M) alone (Fig 1C).

On the other hand, when CB-CD34⁺ cells were treated with puromycin, we could detect CFUs derived CB-CD34⁺ cells only from cells electroporated with the donor matrix plus the *PKLR* TALEN® (Fig 1D). Puro^R cells gave rise to myeloid and erythroid CFUs (Fig 1E). Hematopoietic colonies were individually picked and analyzed by nested PCR to identify the specific integration of the matrix in the *PKLR* locus (see arrows in Fig 1A). We used a nested PCR to increase sensitivity and reduce non-specific amplifications. Properly edited human hematopoietic progenitors showed the specific 2kb PCR band (Fig 2A). Additionally, Sanger sequencing of nested PCR product from colonies confirmed the specific integration in the *PKLR* locus (Fig 2B). Up to 74% of the analyzed CFUs were positive for the knock-in integration (Table 1), showing the high efficiency of the correct integration following the nucleofection and puromycin selection processes. Furthermore, positive CFUs for the knock-in of donor matrix were evaluated to investigate if the specific integration occurred in a single allele (heterozygous Knock-In) or in both alleles (homozygous Knock-In). To address this point, we performed a PCR around target site, which a single 301bp band will detected, amplified from

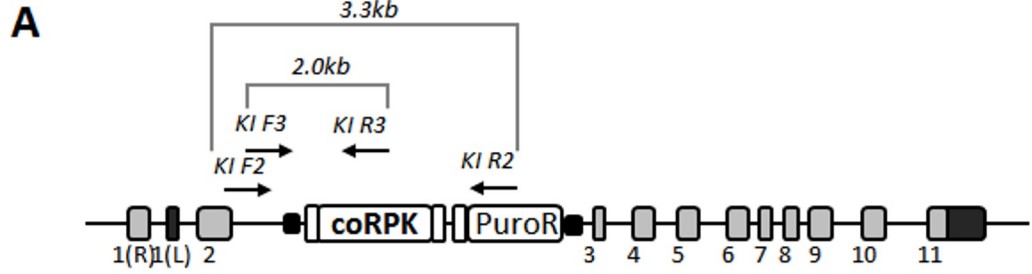

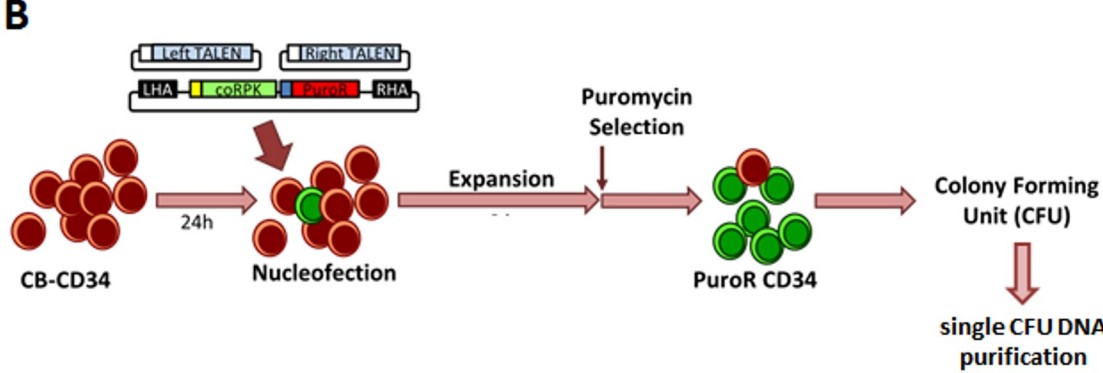

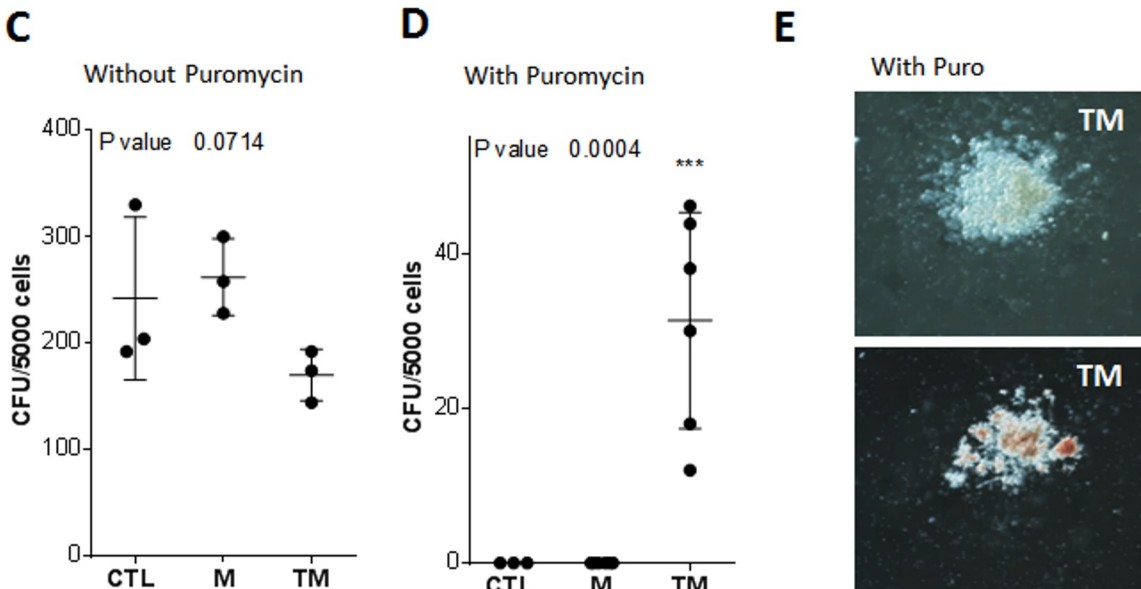

**Fig 1. Gene editing protocol to introduce coRPK cassette in human hematopoietic progenitors.** (A) Scheme of knock-in of donor matrix in *PKLR* locus and the nested PCR analysis applied. (B) Protocol of gene editing of *PKLR* locus in hematopoietic progenitors. Cord Blood Hematopoietic progenitors (CB-CD34) where thawed and pre-stimulated for 24 hours. $1 \times 10^6$ CB-CD34[+] cells were nucleofected with the homologous recombination matrix (M) and *PKLR* specific TALEN[®] (T) targeting a specific sequence in the second intron of *PKLR*. Then, the CB-CD34[+] cells were expanded for 6 days and selected with 1μg/mL of puromycin for 4 additional days. (C) Colony forming ability of control (CTL) human CB-CD34[+] cells, CB-CD34[+] cells nucleofected with the recombination matrix (M) or with *PKLR* TALEN[®] plus recombination matrix (TM) after puromycin selection period. (D) Analysis of CFUs after puromycin selection of TM-nucleofected CB-CD34[+] cells. (E) Images of two representative CFUs derived from TM-nucleofected CB-CD34[+] cells, myeloid CFU (top) and erythroid CFU (bottom). Kruskal-Wallis test was performed; P value is indicated in the figure. (n = 3; mean±SD).

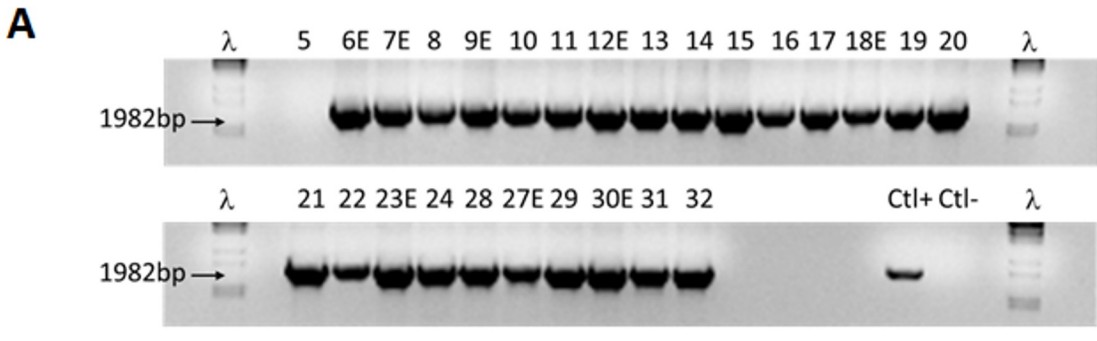

**A**

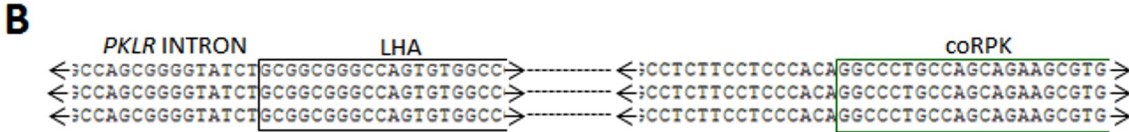

**B**

**Fig 2. Gene editing of the *PKLR* locus in human hematopoietic progenitors.** (A) Analysis of homologous directed repair (HDR) in CFUs. CFUs from Puro^R cells were picked individually. DNA from single CFUs was purified and the HDR was analyzed by nested PCR. This nested PCR was performed using the primers described in Fig 1A. Two sequential PCRs were performed with KI F2 and KI R2 primers and KI F3 and KI R3. Expected 1982bp band was evidenced in an agarose gel. ʎ DNA marker was used to determine the PCR band size. (B) Sanger sequencing of nested PCR products from two different CFUs (bottom sequences) and the theoretical sequence after knock-in (top sequence). Sequences of endogenous second intron, left homology arm and codon optimized RPK cDNA sequences were detected in the Sanger sequence, confirming the specific integration of the recombination matrix.

non-knocked-in allele, but no 301bp band will be detected when both alleles had been knocked-in by the donor matrix (see S1A Fig). The percentage of CFUs with the 301bp band was 53.8% of the total and the rest of the CFUs lacked of this band, which pointed out that the specific integration had occurred in both alleles in 46% of gene edited CFUs (Representative experiment shown in S1B Fig). Despite the high percentage of gene edited hematopoietic progenitors, the overall frequency of gene editing with this approach was 5.8 knocked-in CFUs out of 10,000 in the absence of puromycin selection CFUs which falls short of what is needed for potential clinical application.

In order to enhance the efficiency of the process, we focused on two different aspects of the protocol: the improvement of the gene editing tools and the optimization and shortening of the HSPC ex vivo culture conditions.

## Improvement of *PKLR* gene editing tools

First, we modified the previously developed gene editing tools to reduce their toxicity in HSPCs and enhance the overall gene editing efficacy. We explored the use of *PKLR* TALEN® as mRNA to reduce the toxicity associated to the nucleofection of DNA. To improve the stability of the *PKLR* TALEN® mRNAs, two different modifications were introduced both, to

**Table 1. Homologous directed repair frequency in hematopoietic progenitors.**

| 6d+4d protocol | CFU from Puro^R cells | HR CFU | %HR |
|---|---|---|---|
| CD34-HR1 | 30 | 29 | 96.7 |
| CD34-HR2 | 5 | 2 | 40.0 |
| CD34-HR3 | 21 | 18 | 85.7 |
| | | **Mean** | **74.1** |
| | | SEM | 17.4 |

Data from three independent experiments indicating the number of CFU per 5000 Puro^R cells, the number of CFUs positives for HDR analysis and the percentage of gene edited CFUs. All the CFUs were derived from TM transfected and puromycin selected hematopoietic progenitors. No CFU from either CTL or M nucleofected cells was identified.

stabilize the mRNA by including the 3'UTR of the *β*-Globin gene[31], and to reduce the immune response against exogenous mRNAs by adding the 5'UTR of the Venezuelan Equine Encephalitis Virus (VEEV). The secondary structure of 5'UTR of VEEV, a pathogenic alphavirus, is able to alter *Ifit1* functions, acting as an evasion mechanism to avoid cellular immune response against exogenous mRNAs[30] (Fig 3A).

CB-CD34+ cells were nucleofected with either *PKLR* TALEN® as plasmid DNA or as mRNA with the different modifications (unmodified mRNA, 5'UTR VEEV mRNA and mRNA 3´UTR *β*-Globin). Surveyor analyses were carried out 3 and 7 days after nucleofection (dpn) and/or after CFU assays to determine the efficiency of gene editing. The highest proportion of indels generated by the *PKLR* TALEN® DNA plasmids, which did not have any of these RNA stabilizer domains, was observed at 3 dpn (up to 17%) (Fig 3B). However, the proportion of indels dropped to 0.4% at 7 dpn, pointing to the toxicity associated with DNA nucleofection. On the contrary, nucleofection of 0.25µg or 0.5µg *PKLR* TALEN® as mRNA harboring both mentioned modifications (5'mRNA3') gave rise to 5.4% and 2.8% indels at 3 dpn, respectively. These percentages did not vary considerably at 7dpn (3.0% with 0.25µg and 3.4% with 0.5µg)(S2 Fig). Moreover, the number of colonies generated from CB-CD34+ cells electroporated with *PKLR* TALEN® mRNA was not affected (Fig 3C), indicating the low toxicity of the modified mRNAs nucleofection in CB-CD34+ cells.

Additionally, we explored another nuclease platform, CRISPR/Cas9 system, which was designed to target the same DNA sequence recognized by the *PKLR* TALEN®. A single guide RNA (sgRNA) was identified using Zhang's software (http://crispr.mit.edu/). The selected *PKLR* sgRNA (TAGGGTCTCGTCTGTCACCT) targeted the same sequence as *PKLR* TALEN® did (see Fig 4A). The nucleofection of CB-CD34+ cells with different formulations of the CRISPR/Cas9 system (as all-in-one DNA plasmid, mRNA or Ribonucleoprotein [RNP]) induced indels in a similar manner as the tested TALEN® when analysed by Surveyor assay, although observing a slight increase in the indel generation when introduced as RNP from Supplier A (see Fig 4B and S2 Fig). To analyse more in deep the nuclease activity of *PKLR* CRISPR/Cas9 system delivered as RNP into CB-CD34+ cells in comparison with *PKLR* TALEN® introduced as plasmid DNA, we quantified the INDEL frequency measured by TIDE software in 3 independent experiments. We didn't observe a significant increase in the percentage of indels when using CRISPR/Cas9 system (see Fig 4C).

To test the efficacy of homologous recombination with the different engineered nucleases, we transfected HEK293T cells with the developed donor matrix together with either the *PKLR* TALEN® plasmids, the *PKLR* TALEN® mRNAs or the Cas9-*PKLR* sgRNA all-in one plasmid. Transfected cells were subcloned in the presence of puromycin (Puro), and PuroR clones were analyzed by nested PCR (Fig 4D). Percentages of gene-modified PuroR clones with *PKLR* TALEN® DNA and mRNA were similar: 10 out of 16 and 13 out of 16, respectively. However, the proportion of clones with the specific integration of the donor matrix was lower when cells were transfected with the Cas9-*PKLR* sgRNA all-in-one plasmid. Only 4 out of the 16 PuroR clones were positive for the correct integration, revealing that non-specific integration could also occur with this nuclease platform. These data indicated that our knock-in strategy to insert donor matrix at the second intron is compatible with other nuclease platforms, such as CRISPR-Cas9 system, although the *PKLR* TALEN® RNAs, including the 5' and 3' modifications showed the highest efficiency.

## Improvement of culture conditions for human hematopoietic stem cell gene editing

To improve the maintenance of HSPCs during our gene editing procedure, we first shortened the incubation period after nucleofection from 6 to 4 days. Similarly, the selection process was

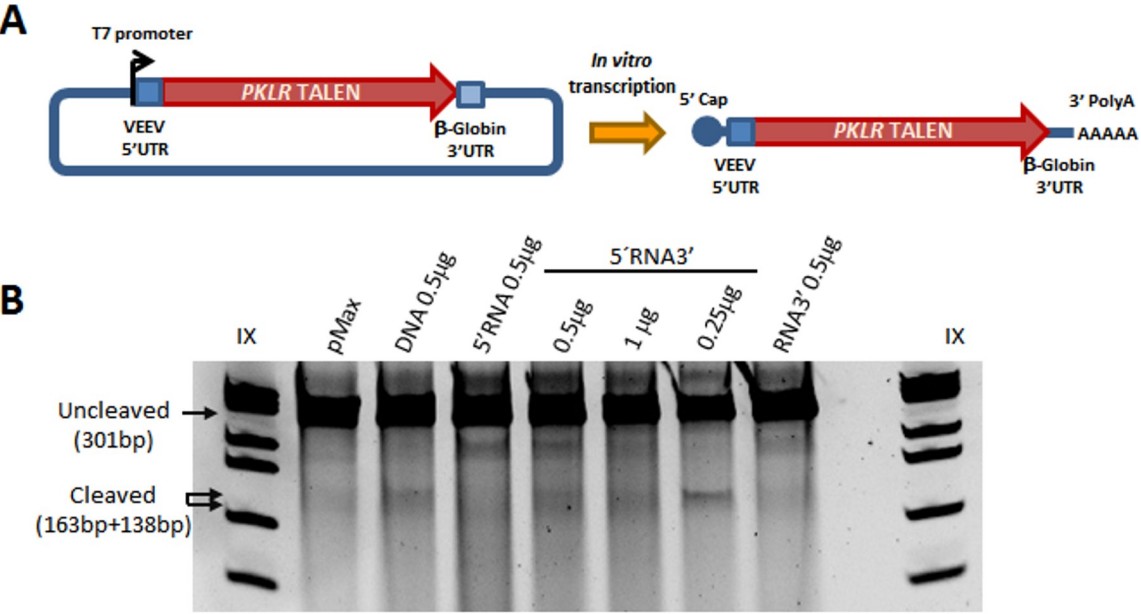

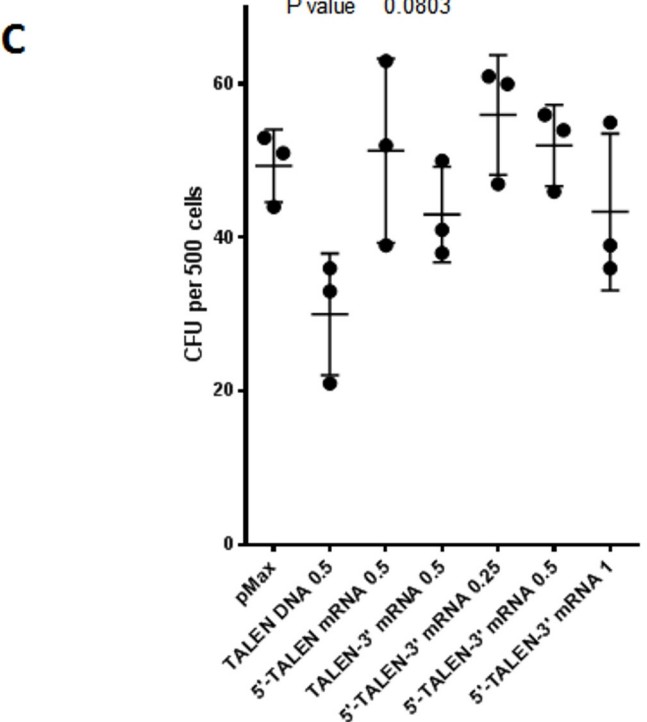

**Fig 3. Gene editing with *PKLR* TALEN® mRNA.** (A) Diagram of 5'UTR of of the Venezuelan Equine Encephalitis Virus (VEEV) (and 3'UTR β-Globin modifications added in *PKLR* TALEN® subunits to synthesis *in vitro* mRNA. (B) Representative Surveyor assay of nucleofected CB-CD34⁺ cells with *PKLR* TALEN® DNA plasmids or different doses of *PKLR* TALEN mRNA with at VEEV 5', β-globin 3' or at both UTRs at 3 days after nucleofection. (C) CFU analysis of nucleofected CB-CD34⁺ with *PKLR* TALEN® DNA plasmids or different doses of mRNA. Kruskal-Wallis test was performed; P value is indicated in the figure. (n = 3; mean±SD).

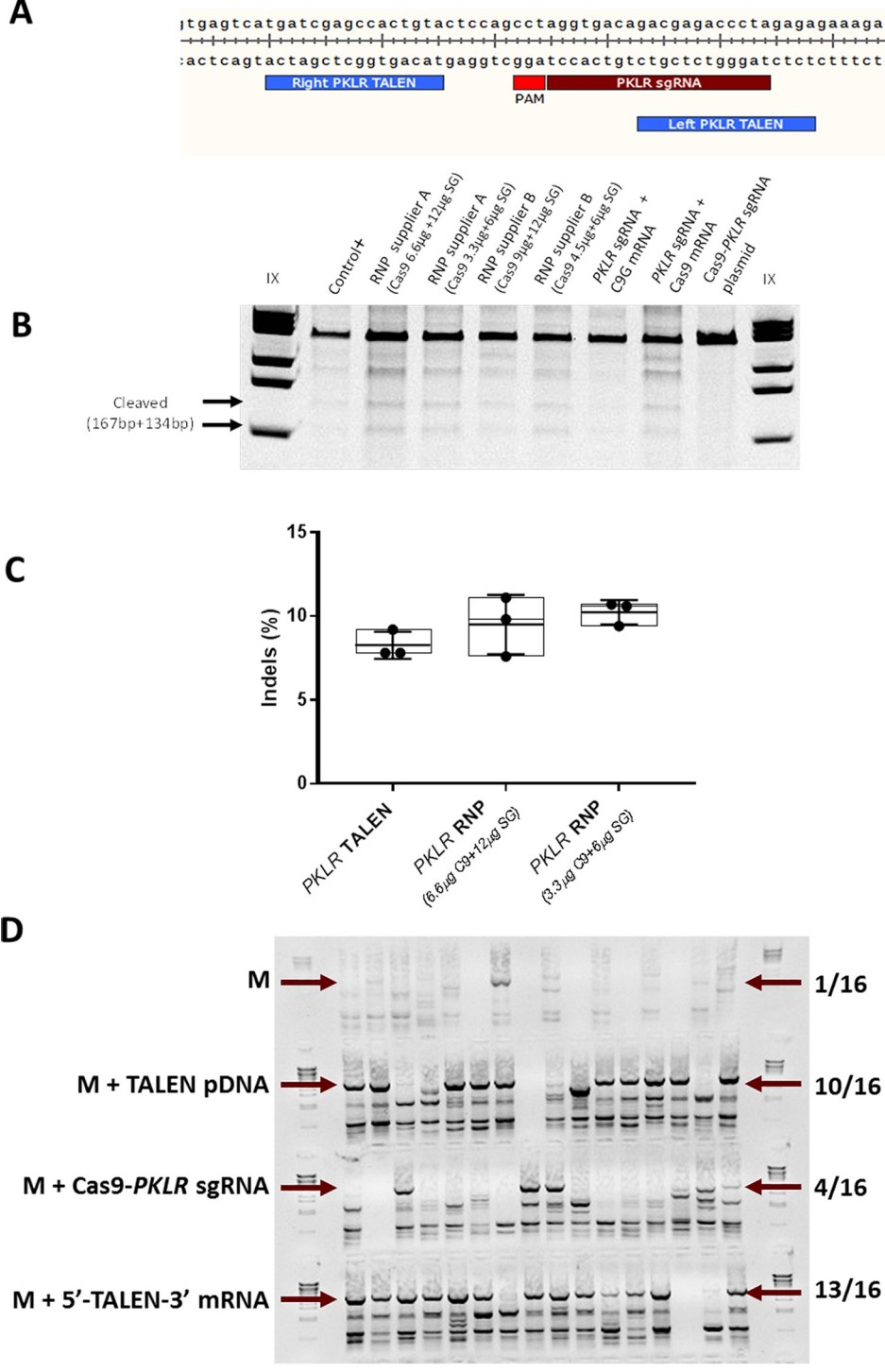

**Fig 4. Gene editing with *PKLR* CRISPR/Cas9 system.** (A) Diagram of targeting site at second intron of *PKLR* gene of the two subunits of *PKLR* TALEN® and *PKLR* sgRNA. (B) Representative Surveyor assay of nucleofected CB-CD34⁺ cells with different formats of CRISPR/Cas9 system specifically targeting *PKLR* locus (Ribonucleoprotein [RNP] assembled with Cas9 protein from either supplier A or B and *PKLR* sgRNA [SG] from IDT according Materials and Methods, in-vitro transcribed Cas9-2A-GFP mRNA [C9G mRNA] plus *PKLR* sgRNA, in-vitro transcribed Cas9 mRNA plas *PKLR* sgRNA, all-in-on plasmid [Cas9-*PKLR* sgRNA plasmid]). Used amount of each reagent is indicated. (C) Indels quantification measured by TIDE analysis. Comparison between CB-CD34⁺ cells nucleofected with *PKLR* TALEN® introduced as plasmid DNA (8.26% indels) and different concentrations of RNP from supplier A (9.5% and 10.2% indels). (D) Nested PCR analysis, as previously described, to identify gene editing in HEK293T clones after

being transfected with the donor matrix plus *PKLR* TALEN® DNA plasmids, *PKLR* TALEN® mRNA or *PKLR* sgRNA CRISPR/Cas9 plasmid. The expected band size is marked with an arrow. λ DNA marker was used to identify the PCR band size.

shortened from 4 to 2 days (4d+2d protocol). As shown in Table 2, the percentage of gene-edited CFUs did not change significantly compared to the previous longer protocol (67% of CFUs were positive for the specific integration [Table 2]).

In order to test how efficient these new conditions were to gene edit human HSCs capable of engrafting irradiated imnodeficient NSG (NOD.Cg-Prkdc$^{scid}$Il2rg$^{tm1Wjl}$/SzJ) mice. As starting population, $1x10^6$ CB-CD34$^+$ cells per animal were nucleofected with the donor matrix (M) plus the *PKLR* TALEN®, as DNA plasmids (2 different experiments, 4 mice in total) or mRNAs (2 different experiments, 4 mice in total) with the stabilizer modifications already described, using the 4+2 editing protocol (Fig 5A). As result of the nucleofection and the puromycin selection, from $8.4x10^4$ to $1.35x10^5$ surviving Puro$^R$ cells were transplanted per NSG mouse. When PKLR TALEN® were electroporated as DNA, a very low human engraftment (0.78%) was detected in the animal transplanted with the highest number of Puro$^R$ cells, and no human engraftment was detected in the other animals animals. No HR could be detected in the engrafted human cells. On the contrary, a better human engraftment (5.57%) was present in one out of the four animals transplanted with cells electroporated with M and *PKLR* TALEN® mRNA (Fig 5B). Again, no HR was detected in total hCD45$^+$ cells sorted from the NSG mouse bone marrow. Among human hematopoietic cells, 1.8% were hCD45$^+$hCD34$^+$. This human progenitor's population was purified, re-selected with puromycin, and cultured in methylcellulose to investigate the presence of knock-in integrations in human hematopoietic progenitors present in the mouse 90 days post-transplant. One out of the 27 colonies derived from engrafted human CD34$^+$ cells was positive for HDR (Fig 5C), evidencing the *PKLR* gene editing in human HSCs, although still at a very low efficacy.

16,16-Dimethyl Prostaglandin E2 (dmPGE$_2$) has been previously reported to increase homing, survival and proliferation of hematopoietic progenitors and to enhance human hematopoietic progenitor engraftment[33, 34]. In addition, dmPGE$_2$ has been also observed to improve gene targeting in HSCs[18]. Therefore, we investigated the direct effect of dmPGE$_2$ in the gene editing of engraftable HSPCs. CB-CD34$^+$ cells were nucleofected with the donor matrix (M) plus the *PKLR* TALEN® as DNA plasmids or as mRNA. One million nucleofected cells were drug selected using the 4d+2d protocol, as described above, in the presence of dmPGE$_2$. $3.3x10^4$ Puro$^R$ HSPCs derived of the M and *PKLR* TALEN DNA plasmid nucleofection and $1.1x10^5$ Puro$^R$ cells from the M and *PKLR* TALEN mRNA nucleofection were then

**Table 2. Gene editing frequency in hematopoietic progenitors.**

| 4d+2d protocol | CFU from Puro$^R$ cells | HR CFU | %HR |
|---|---|---|---|
| CD34-HR4 | 6 | 6 | 100 |
| CD34-HR5 | 21 | 15 | 71.4 |
| CD34-HR6 | 40 | 11 | 27.5 |
| CD34-HR7 | 21 | 15 | 71.4 |
| | | **Mean** | **67.6** |
| | | SEM | 15.0 |

Data from four independent experiments indicating the number of CFUs per 5000 Puro$^R$ cells, the number of CFUs positives for homologous recombination analysis and the percentage of gene edited CFUs. All the CFUs were derived from TM transfected and puromycin selected hematopoietic progenitors.

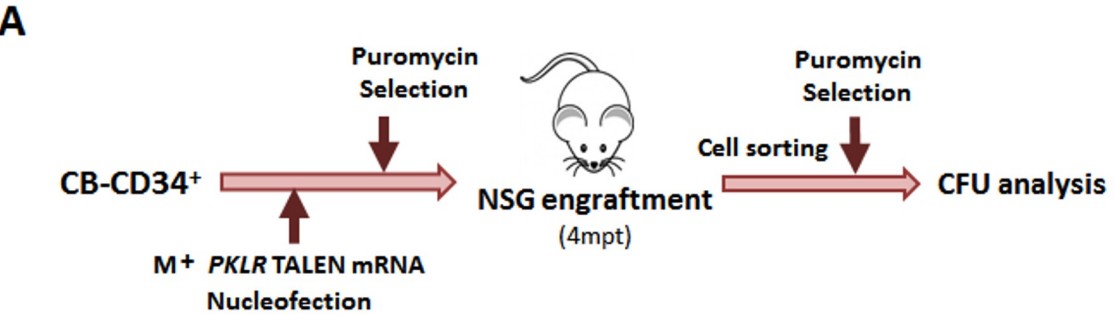

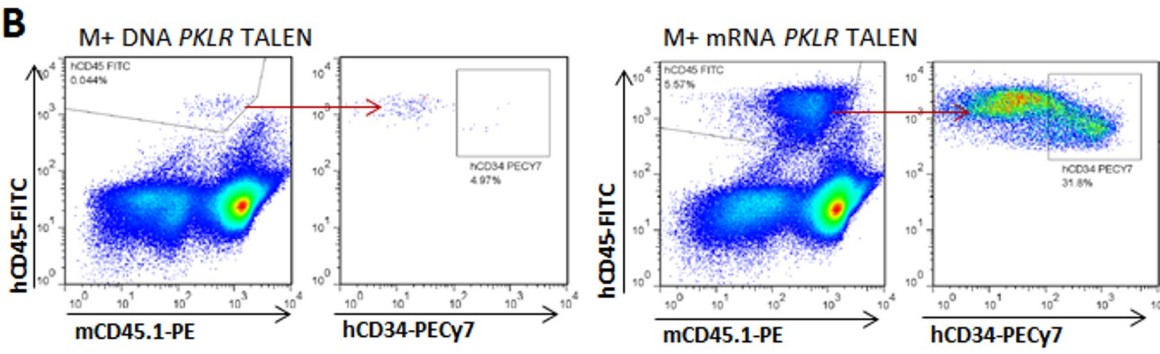

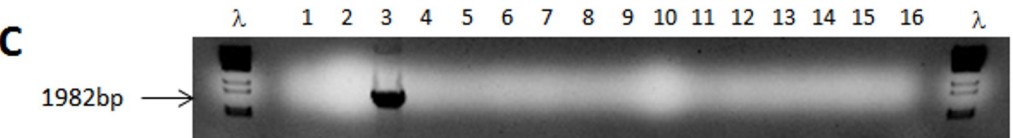

**Fig 5. Engraftment of gene editing human hematopoietic progenitors in immunodeficient mice.** (A) Diagram to analyze gene editing in human HPSCs engrafted in NSG mice. Fresh CB-CD34$^+$ cells were electroporated by the donor matrix plus *PKLR* TALEN$^®$, either as plasmid DNA or mRNA. After shortening culture, up to 4 days, and puromycin selection time, up to 2 days, gene editing was analyzed by nested PCR in CFUs and in engrafted human hematopoietic progenitors in NSG mice, after enrichment with for. (B) FACS analysis of human engraftment in the bone marrow of NSG mice 4 month post-transplant. Human hematopoietic population (hCD45$^+$) is shown as well hCD34$^+$ within the human population. Representative FACS analysis of an animal transplanted with CB-CD34$^+$ cells electroporated with *PKLR* TALEN$^®$ DNA plasmids (left side) or *PKLR* TALEN$^®$ mRNAs (right side). (C) Nested PCR analysis of CFU derived of hCD45$^+$hCD34$^+$ sorted from mouse bone marrow and selected by puromycin. The expected band size is marked with an arrow. λ DNA marker was used to identify the PCR band size.

transplanted intravenously into four irradiated NSG mice (2 experiments, 4 mice in total). High levels of human hematopoietic cells were detected 4 months after the infusion of nucleofected cells treated with dmPGE$_2$ (Fig 6A). However, no specific integration was observed in hCD45$^+$ cells from the BM of these mice. Moreover, hCD34$^+$ cells from the human hCD45$^+$ population were sorted and re-selected with puromycin for 2 days prior to conducting CFU assays. None of these CFUs derived from the engrafted human progenitors had integrated the specific sequence. To verify the impact of dmPGE$_2$ on our gene editing approach, CB-CD34$^+$ cells were nucleofected with the donor matrix (M) plus the *PKLR* TALEN$^®$ as plasmid DNA,

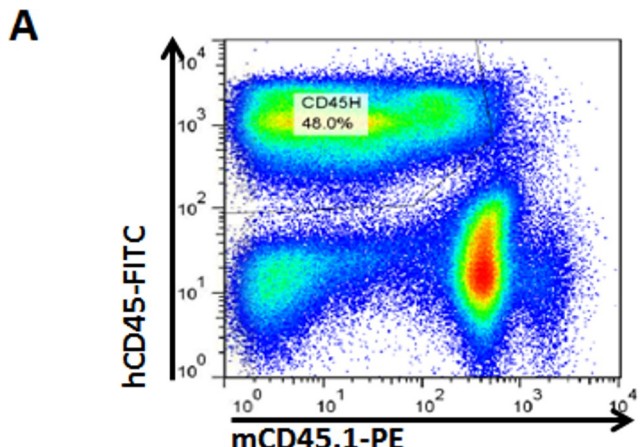

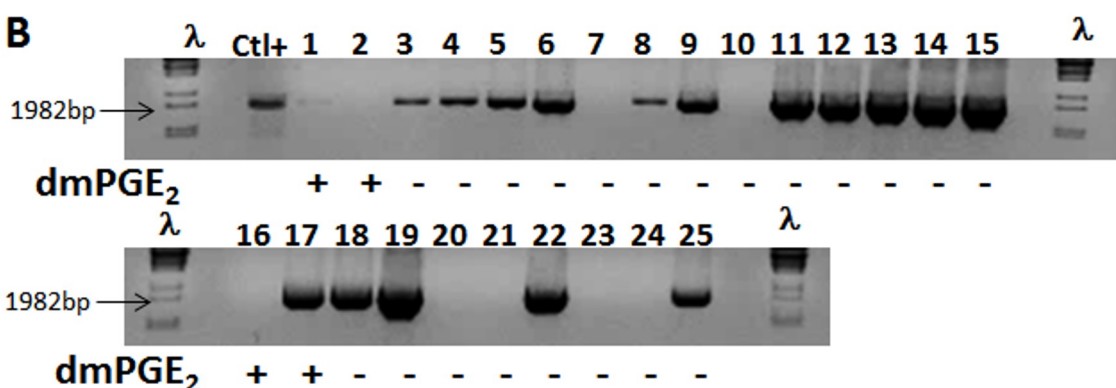

**Fig 6. Effect of dmPGE₂ in the gene editing protocol.** (A) Representative FACS analysis of human engraftment 4 months post-transplant of NSG mice transplanted with nucleofected CB-CD34⁺ cells with *PKLR* TALEN® and donor matrix and cultured in presence of dmPGE₂. (B) Nested PCR analysis of individual CFUs derived from puromycin resistant CB-CD34⁺ cells after *PKLR* TALEN® and donor matrix nucleofection. Presence or absence of dmPGE₂ is indicated. The expected band size is marked with an arrow. λ DNA marker was used to identify the PCR band size.

puromycin-selected in presence or absence of dmPGE₂, and seeded in methylcellulose. We observed fewer colonies when the hCD34⁺ cells had been treated with dmPGE₂; and more interestingly, the percentage of CFUs with the specific integration were lower when dmPGE₂ was present than without this drug (Table 3 and Fig 6B). These results indicated that the use of dmPGE₂ in our gene editing strategy did not improve the efficiency of the protocol.

Altogether, our results show that the use of TALEN as mRNAs with 5' and 3' modifications and the reduction in the protocol duration facilitate the overall efficiency of the gene editing of

**Table 3. Gene editing frequency with/without dmPGE₂.**

| Condition | %HR in PuroR CFU |
| --- | --- |
| -dmPGE₂ | 71.4% (15/21) |
| +dmPGE₂ | 25% (1/4) |

CB-CD34⁺ cells were edited with the usual protocol. Part of the culture was treated with dmPGE₂ and seeded in methylcellulose. The number of colonies obtained from puromycin resistant CB-CD34+ cells cultured with dmPGE2 was very low in comparison with the untreated cells, and the percentage of HDR was decreased as well.

hCD34[+] cells in the *PKLR* locus and point out the feasibility of the presented knock-in strategy to correct *PKLR* mutations. However, the efficiency and yield of corrected cells need further improvement in order to be clinically applicable.

## Discussion

In the present work, a knock-in gene editing approach previously described to correct PKD in patient-specific iPSC[17] has been adapted to HSPCs. After nucleofection and puromycin selection, up to 70% of HSPCs showed specific integration *in vitro*, although a marked cellular toxicity was associated with the procedure. To reduce toxicity while increasing efficacy, we strove to find the most suitable gene editing tools and define optimal expansion and selection timing. Using improved gene editing tools and shortening the former protocol, we were able to maintain a high level of gene editing in hematopoietic progenitors. However, gene edited HSPCs could be detected after engraftment in NSG mice, although with a very low efficiency. Altogether, the presented knock-in approach is feasible to gene modify hematopoietic progenitors at the second intron of the *PKLR* gene.

We and others have demonstrated correction of inherited hematopoietic diseases through gene editing in iPSC[10–13, 15–17, 36]. Although corrected iPSCs can be therapeutic alternatives for some pathologies[37], they are not a clinically applicable cell source for the treatment of blood diseases yet. Different research groups have made significant improvements in the generation of HSPCs from iPSC[38–40]. However, this process is still very inefficient. Therefore, important efforts have been made to apply gene editing technology directly to HSPCs, overcoming the important difficulties associated with the hematopoietic differentiation from iPSCs. So far, only few groups have described a level of gene editing efficacy in HSPCs potentially compatible with their clinical use[18, 24, 25, 41]. In those reports, a combination of nucleases provided as DNA, mRNA or RNP together with viral vectors carrying the therapeutic matrix allowed gene modification in human CD34[+] cells.

Many efforts have been made in order to improve endonuclease activity and delivery into target cells. Nucleofection of TALEN mRNA clearly reduced the toxicity of the protocol. On the other hand, it has been demonstrated that 5' and 3' modifications of messenger RNAs could improve the stability of the mRNA and their overall translation efficiency. Some of this modifications include the 3'UTR of β-Globin, which has demonstrated improvement in mRNA stability[31] and the 5'UTR secondary structure of the VEEV, which has been shown to alter *Ifit1* binding and function as an evasion mechanism by which alphaviruses use RNA structural motifs to avoid immune restriction[30]. Here, we have observed that TALEN mRNAs with both modifications facilitate the use of this nuclease in human HPSCs.

Here, we combined TALEN mRNA and donor matrix as plasmid DNA to get efficient specific integration of the therapeutic donor in the desired locus in CD34[+] cells. Up to 70% of TALEN-treated HSPCs were edited after puromycin selection. We believe that this efficacy of gene correction would be sufficient to correct PKD after transplantation. Indeed, a minimum of 30% corrected stem cells, either healthy or transduced with retroviral vectors, is sufficient to compensate the diseased phenotype[42]. However, the amount of corrected HSCs with the ability to engraft NSG mice obtained and the overall yield of the protocol is still too low to be considered as clinically applicable.

Some limitations can be seen that diminish the overall effectiveness of the protocol. First, nucleases should be active at the appropriate moment to facilitate gene editing. Thus, co-nucleofection of nucleases and repair matrix may not be the optimal solution. Second, the delivery methods to provide the repair matrix may impact the overall gene editing efficacy. The nucleofection procedure has been shown to be very toxic in some cases[43, 44]. We have

observed similar toxic effects here when applied to *PKLR* gene editing in CD34$^+$ cells. Efforts to tackle this limitation by the use of microfluids[45] or vectorizing the donor therapeutic matrix using AAV viral vectors have been shown to be very efficient in the case of the β-Globin gene[23]. Both improvements are being currently explored in our laboratory. Third, puromycin selection system might favor the selection of hematopoietic progenitors with several integrations. Indeed, there was a considerably number of Puro$^R$ CFUs without the specific integration; those clones either can have integrated the donor matrix in different genomic sites randomly, or may have not been properly selected with the treatment. It should be noted, however, that the percentage of hematopoietic progenitors with integration in the two alleles of *PKLR* locus is high (S1 Fig), which might reflect the selective pressure of the puromycin treatment.

All these improvements aim to get a clinically relevant gene editing efficacy in primitive HSCs. Although we were able to obtain a significant frequency of gene edited hematopoietic progenitors, we failed to target the most primitive HPSCs, getting only a small evidence of gene editing in NSG engraftable HSCs.

CRISPR/Cas9 system is nowadays frequently used because of its convenience for design and implementation[46, 47]. Our results indicate that, at least in this genomic region, the developed TALEN and the CRISPR/Cas9 nucleases are performing similarly in terms of cleavage efficiency (see Fig 4B and 4C and S2 Fig). However, *PKLR* TALEN$^®$ showed a slightly higher efficacy triggering targeted insertion of therapeutic donor matrix. Recently, CRISPR/Cas9 nuclease has been reported to be more efficient than TALEN[41, 48] for genome editing. This apparent discrepancy could be explained by (i) differences in the target sgRNA used in these studies, or (ii) by TALEN design its self. The 5' and 3' modifications introduced within our TALEN mRNAs could also favor the better performance of the specific TALE nucleases. Additionally, the use of CRISPR/Cas9 nuclease as RNP is known to improve the efficacy of cutting, which was also verified in our system, but this *PKLR* RNP efficacy was similar to the *PKLR* TALEN mRNA (Fig 4B and 4C and S2 Fig), so we can conclude that both platforms, *PKLR* TALEN and *PKLR* RNP, have similar efficacy to target the second intron of *PKLR* gene.

A very important factor in achieving efficient gene editing in HSPC is the availability of optimal *in vitro* culture conditions that facilitates their genetic manipulation without losing their stem cell properties. Many efforts and improvements have been made over the last decades in this respect. However, up to now it has only been possible to maintain HSPC *in vitro* for a very limited time[49, 50]. In this report, we have reduced the time required for the manipulation and selection of the cells without losing gene editing efficacy. Unexpectedly, although dmPGE$_2$ has recently been reported as a very useful molecule to enhance HPSC survival, proliferation and engraftment[51, 52], and to help in HPSC gene editing[18], in our hands the addition of dmPGE$_2$ to our culture conditions decreased the efficiency to enrich gene editing HSPCs. Perhaps its beneficial effect increased the resistance of HSPCs (gene edited or not) to the puromycin selection, consequently human engraftment was enhanced since the survival of HSPCs without HDR was promoted by dmPGE$_2$ treatment.

Overall, we show that gene editing of the *PKLR* locus in HSPCs is feasible but it might be applied in the clinical setting only after further improvements aimed at increasing efficiency, yield and targeting of HSCs. The field of gene editing is developing very fast. New improvements combined with those described in this manuscript will make the correction of *PKLR* gene by gene editing clinically applicable for the treatment of PKD in the near future.

## Supporting information

**S1 Fig. Monoallelic and biallelic gene editing frequencies.** (A) Diagram of PCR analysis. Positive CFU for nested PCR were re-evaluated to test if the integration of our matrix had

occurred in one allele or in both alleles. (B) Representative data from one of the independent experiments shown the analysis of monoallelic/biallelic gene editing of *PKLR* locus in human hematopoietic progenitors. Positive bands meant single allele integration, and no band indicated biallelic integration.
(PDF)

**S2 Fig. Quantification of Indel percentage at *PKLR* gene of nucleofected CB-CD34⁺ cells with different format of *PKLR* TALEN and CRISPR/Cas9 at 7dpn.** Used amount of each reagent is indicated. Data from four independent experiments.
(PDF)

## Acknowledgments

The authors would like to thank Miguel A. Martin for the careful maintenance of NSG mice, and Rebeca Sánchez and Omaira Alberquilla for their technical assistance in flow cytometry. The authors also thank Fundación Botín for promoting translational research at the Hematopoietic Innovative Therapies Division of the CIEMAT.

## Author Contributions

**Conceptualization:** Oscar Quintana-Bustamante, Jose C. Segovia.

**Formal analysis:** Oscar Quintana-Bustamante, Sara Fañanas-Baquero, Jose C. Segovia.

**Funding acquisition:** Juan A. Bueren, Jose C. Segovia.

**Investigation:** Oscar Quintana-Bustamante, Sara Fañanas-Baquero, Jose C. Segovia.

**Methodology:** Oscar Quintana-Bustamante, Sara Fañanas-Baquero, Israel Orman.

**Project administration:** Oscar Quintana-Bustamante, Jose C. Segovia.

**Resources:** Oscar Quintana-Bustamante, Raul Torres, Philippe Duchateau, Laurent Poirot, Agnès Gouble.

**Supervision:** Oscar Quintana-Bustamante, Jose C. Segovia.

**Validation:** Sara Fañanas-Baquero, Jose C. Segovia.

**Visualization:** Oscar Quintana-Bustamante, Sara Fañanas-Baquero.

**Writing – original draft:** Oscar Quintana-Bustamante, Sara Fañanas-Baquero, Jose C. Segovia.

**Writing – review & editing:** Oscar Quintana-Bustamante, Sara Fañanas-Baquero, Juan A. Bueren, Jose C. Segovia.

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
