## [Decision Letter · Decision Letter 0]

9 Jul 2019

PONE-D-19-14838

Gene editing of PKLR gene in human hematopoietic progenitors through 5’ and 3’ UTR modified TALEN mRNA

PLOS ONE

Dear Mr Quintana Bustamante,

Thank you for submitting your manuscript to PLOS ONE. After careful consideration, we feel that it has merit but does not fully meet PLOS ONE’s publication criteria as it currently stands. Therefore, we invite you to submit a revised version of the manuscript that addresses the points raised during the review process.

Please address all points of the reviewers, with a particular emphasis on technical accuracy because this is the main PLoS ONE editorial criterion. Both reviewers raise some technical issues which appear to be mostly addressable with written corrections, except for the DNA sequence analysis which may require further work.

We would appreciate receiving your revised manuscript by Aug 23 2019 11:59PM. To enhance the reproducibility of your results, we recommend that if applicable you deposit your laboratory protocols in protocols.io, where a protocol can be assigned its own identifier (DOI) such that it can be cited independently in the future. For instructions see: http://journals.plos.org/plosone/s/submission-guidelines#loc-laboratory-protocols

We look forward to receiving your revised manuscript.

Kind regards,

Mark Isalan

Academic Editor

PLOS ONE

**Journal Requirements:**

2. Thank you for stating that “The funders had no role in study design, data collection and analysis, decision to publish, or preparation of the manuscript” in your financial disclosure.

Please also provide the name of the funders of this study (as well as grant numbers if available) in your financial disclosure statement.

"PD, LP and AG are employee of Cellectis.JAB and JCS are consultants for Rocket Pharmaceuticals. "

**Comments to the Author**

1. Is the manuscript technically sound, and do the data support the conclusions?

Reviewer #1: Yes

Reviewer #2: Partly

2. Has the statistical analysis been performed appropriately and rigorously? 

Reviewer #1: No

Reviewer #2: Yes

3. Have the authors made all data underlying the findings in their manuscript fully available?

Reviewer #1: Yes

Reviewer #2: Yes

4. Is the manuscript presented in an intelligible fashion and written in standard English?

Reviewer #1: Yes

Reviewer #2: Yes

5. Review Comments to the Author

Reviewer #1: This manuscript reported a feasible way, although inefficient, to edit PKLR gene in HPSCs, which may be clinically applicable to correct PKD in the future after further improvements. Through a thorough reading and reviewing, I think this manuscript can be accepted after minor revision. The main questions in this manuscript to my point of view are as follows.

1. Abbreviations (e.g. RPK and PKLR) should be explained at the first occurrence in the text.

2.

3. According to the supplementary data, several figures in the first paragraph of page 13 seem to be miscalculated: 56% should be 53.8%, while 44% should be 46%. In addition, the source of the data for “5.8 knock-in CFUs out of 1000 initial CBCD34+ cells” could not be found.

4.

5. In Results part, the authors declare that the use of dmPGE2 did not improve the efficiency of the specific integration. However it seems cursory to draw this conclusion just based on 4 CFUs. To support this conclusion, more puromycin-selected colonies are needs for further analysis although the dmPGE2 treatment will do harm to hCD34+ cells and make it hard to yield colonies.

6. The figure legends in the text are too simple to understand. It is important to provide enough information so that the readers can easily understand the content of article. For example, what does the λ in Figure 2A and IX in Figure 3B mean? Are they DNA markers or something else?

Reviewer #2: Overview

This manuscript outlines a study using CRISPR- and TALEN-directed gene editing to repair mutations in the PKLR gene, which produces a pyruvate kinase, mutations within which are responsible for pyruvate kinase deficiency (PKD). This rare metabolic disease has been a target of previous gene editing protocols but the overall outcome was was disappointing sense a low yield of corrected cells was attained. As has been seen for other gene editing protocols, attempts to increase the gene repair efficiency led to a significant level of toxicity. The authors now extend the work by comparing several programmable nucleases and their associated systems coupled to a more clear definition of expansion and selection time. The results are somewhat disappointing but not truly unexpected, since many ex vivo approaches, designed with the best of intentions, lead to a very low level of gene edited HPSCs following engraftment. The authors conclude that they have demonstrated the feasibility of gene editing of the PKLR gene yet further improvements must be made in order to achieve clinical application.

Critique

Overall, this is a standard targeting approach to a metabolic disease, a group of diseases that have a myriad of mutations possibly approachable by gene editing. What sounds so simple is so hard because it's not just about the gene editing, which likely works with high efficiency and specificity now, but rather about the treatment of the cells before and after the reaction. These are well-established authors who have a good track record of publishing solid work and this is no exception. I would suggest some modifications to the manuscript particularly in terms of detail. I think it's also important that the authors develop a stronger argument as to why their work is worth publishing even though for the overall scientific community, they do provide some important information. Just showing that gene editing in PKLR is feasible in a new cell type, is really not meeting the high bar now set for publication in significant journals. My recommendation is to accept the paper with major revision following the points below:

On page 5, in the first section of the Methods and Materials, the authors indicate that they are growing CD34+ cells in antibiotics. While it's clear that using puromycin as a selection is helpful experimentally , this will be problematic in a transition to ex vivo gene editing protocols. Antibiotic treatment of any cells causes major redistribution of metabolic pathways including that associated with gene editing. Why did they choose to do so as it is not usually a good idea? A strong argument needs to be made as to how this would impact clinical application only because their previous work have moved along this path already. This is important points of the will likely affect the outcome of their experiments and the generalization of their experiments into clinical application. It's also important to define the concentration of puromycin in more detail.

The RNP is a logical choice for use in any sort of ex vivo application but it appears out of nowhere and so what concentrations were used and why were they chosen? The RNP has a rapid turnover in the cell and in the nucleus and in some cases does not actually reach the nucleus.

While I do not follow this exact field, I did notice several papers appearing Molecular Therapy (NA) last year that detailed the challenges with delivery of ex vivo therapy (Modarai et al). I often utilize this paper as an example of how the details are important and that transfection efficiency can often modify/slant outcomes... I would reference this paper and discuss how the authors overcome this important experimental parameter.

What is the role of dmPGE(2)? ( page 7).

While this is not the authors problem, it's now widely accepted that the Surveyor Assay does not sufficiently reflect CRISPR activity in human cells. And, while the equation used to develop the percentage of cleavage is useful, this assay misses small indels less than 3 to 4 bases.... DNA sequence analysis and TIDE must be used in parallel even in the early sections of the work.

The authors need to explain why they do not see any heterozygotes ? I'm actually a bit surprised by this as one would expect to see a small percentage of them.

On page 13, the authors loosely use the term homologous recombination.... It is now true that gene editing has become a garbage heap of terms, mostly inappropriate, the authors need to be quite careful in using the term homologous recombination. The basket term is now considered homology-directed repair since homologous recombination is reserved for chromosomal crossovers that occurr during meiosis and, less often, in mitosis. It's not a big point, but it's a bit sloppy.

I'm a little surprised about the toxicity associated with DNA nucleofection since much of the rest of the world believes that nucleofection is much gentler than electroporation.... Please explain.

On page 16, this goes back to my point about referencing previous data that has already established..i.e. lack of activity in expression constructs as opposed to the RNP. While it is true that senior investigators such as this group have made important contributions and often reflect on their own work, there's no use reinventing the wheel. I think of a simple scan of the literature might be helpful to avoid experimental activity whose outcome is quite predictable, and has been previously reported.

Page 17, once again, the trend outlined in Table 2 has been previously reported... And needs to be cited.

A good place to talk about the delivery variations is in the Disucssion... Previous information, I suggest on page 23 at the bottom continuing on to page 24.

6. PLOS authors have the option to publish the peer review history of their article (what does this mean?). If published, this will include your full peer review and any attached files.

Reviewer #1: No

Reviewer #2: No

---

## [Author Response · Author response to Decision Letter 0]

29 Aug 2019

I have attached a file named "Quintana_Bustamante_al_Response to Reviewers" in which I respond all the comments.

---

## [Decision Letter · Decision Letter 1]

30 Sep 2019

Gene editing of PKLR gene in human hematopoietic progenitors through 5’ and 3’ UTR modified TALEN mRNA

PONE-D-19-14838R1

Dear Dr. Quintana Bustamante,

We are pleased to inform you that your manuscript has been judged scientifically suitable for publication and will be formally accepted for publication once it complies with all outstanding technical requirements.

With kind regards,

Mark Isalan

Section Editor

PLOS ONE

Additional Editor Comments (optional):

Reviewers' comments:

Reviewer's Responses to Questions

**Comments to the Author**

1. If the authors have adequately addressed your comments raised in a previous round of review and you feel that this manuscript is now acceptable for publication, you may indicate that here to bypass the “Comments to the Author” section, enter your conflict of interest statement in the “Confidential to Editor” section, and submit your "Accept" recommendation.

Reviewer #1: All comments have been addressed

Reviewer #2: All comments have been addressed

2. Is the manuscript technically sound, and do the data support the conclusions?

Reviewer #1: Yes

Reviewer #2: Yes

3. Has the statistical analysis been performed appropriately and rigorously? 

Reviewer #1: Yes

Reviewer #2: N/A

4. Have the authors made all data underlying the findings in their manuscript fully available?

Reviewer #1: Yes

Reviewer #2: Yes

5. Is the manuscript presented in an intelligible fashion and written in standard English?

Reviewer #1: Yes

Reviewer #2: Yes

6. Review Comments to the Author

Reviewer #1: Dear Authors, thank you for answering my questions carefully and your answers are in place. I think your article can now be accepted by PLOS ONE. Congratulations

Reviewer #2: The authors have adequately addressed all of the returns that I have. I suggest acceptance of the manuscript as it now stands.

7. PLOS authors have the option to publish the peer review history of their article (what does this mean?). If published, this will include your full peer review and any attached files.

Reviewer #1: No

Reviewer #2: No

---

## [Editor Report · Acceptance letter]

4 Oct 2019

PONE-D-19-14838R1 

Gene editing of *PKLR* gene in human hematopoietic progenitors through 5’ and 3’ UTR modified TALEN mRNA 

Dear Dr. Quintana-Bustamante:

I am pleased to inform you that your manuscript has been deemed suitable for publication in PLOS ONE. Congratulations! Your manuscript is now with our production department. 

With kind regards,

on behalf of

Dr. Mark Isalan 

Section Editor

PLOS ONE